# Risk and prognosis of second primary malignancies in patients with follicular lymphoma in the era of rituximab: A population study based on the SEER database

Ying Tian[1¤a], Wanxi Yang[1¤a], Juan Xu[1¤a], Yuanxiao Li[2¤b], Wenjiao Tang[1¤a], Caigang Xu[1¤a]*

1 Department of Hematology, Institute of Hematology, West China Hospital, Sichuan University, Chengdu, China, 2 Department of Pediatric Gastroenterology, Lanzhou University Second Hospital, Lanzhou, China

¤a Current Address: No. 37 Guoxue Xiang, District Wuhou, Chengdu, Sichuan, China
¤b Current Address: 82 Cuiyingmen Street, Chengguan District, Lanzhou, Gansu Province, China
* xucaigang@wchscu.cn

## Abstract

### Objective

Follicular lymphoma (FL) patients have achieved favorable long-term survival since the introduction of rituximab. However, the development of second primary malignancies (SPMs) indicates a poor survival prognosis for FL patients, and large-scale studies in this field remain limited. This study investigates the prognostic factors for FL patients in the rituximab era, as well as the clinical characteristics, risk factors, and prognosis for patients who developed SPMs.

### Methods

From 2000 to 2020, a total of 33,104 patients with pathologically confirmed FL were identified within the Surveillance, Epidemiology, and End Results (SEER) database. Competing-risk regression analysis was used to assess prognostic factors for lymphoma-specific survival (LSS), risk factors for developing SPMs, and prognosis in FL patients.

### Results

Multivariate analysis identified age ≥ 40 years, Black race, unmarried status, non-urban residence, nodal lymphoma presentation, Grade 3 histology, advanced Ann Arbor stage, and B symptoms as independent adverse prognostic factors for both overall survival (OS) and LSS. Chemotherapy as initial treatment was associated with inferior LSS in FL patients. Protective factors for OS and LSS included female sex, higher income, diagnosis post-2005, diagnosis-to-treatment intervals >1 month, and receipt of radiotherapy or surgery. SPMs correlated with reduced LSS risk in FL

**Data availability statement:** All relevant data are within the manuscript and its Supporting Information files.

**Funding:** Caigang Xu received funding from Sichuan Science and Technology Department for Key Research and Development Program (2019YFS0027).The funders had no role in study design, data collection and analysis, decision to publish, or preparation of the manuscript.

**Competing interests:** The authors have declared that no competing interests exist.

patients. Elevated SPM incidence among patients aged>40years, and non-Hispanic ethnicity, while reduced SPM risks were observed in females, unmarried patients, those receiving non-radiotherapy initial treatment, Grade 3 cases, and patients diagnosed during 2015–2019. Notably, FL patients aged >60 years, unmarried, and those diagnosed post-2010 demonstrated heightened OS risk following SPM development. Conversely, initial radiotherapy conferred protective effects against both OS and LSS in patients with SPMs.

## Conclusion

In this study, we conducted a large, population-based analysis across the United States to identify risk factors for the development of SPMs and to delineate prognostic indicators for FL patients in the context of rituximab therapy, along with the clinical characteristics, risk factors, and prognostic features associated with SPMs. These findings have translational implications for risk-adapted surveillance. Future studies should validate predictive models across diverse healthcare settings, elucidate molecular mechanisms of SPM pathogenesis in FL, and evaluate targeted screening interventions through prospective trials.

## 1. Introduction

Follicular lymphoma (FL), originating from the germinal center B cells in lymph nodes or lymphoid tissue, is one of the most common indolent forms of non-Hodgkin's lymphoma (NHL). In Western countries, FL accounts for about 20–35% of NHL cases [1,2], with an age-standardized incidence of 2–4 per 100,000 person-years [1]. The therapeutic landscape for FL has been revolutionized by anti-CD20 monoclonal antibodies, particularly rituximab-based immunochemotherapy, which has elevated 10-year overall survival (OS) rates beyond 80% [2]. While these advancements have substantially improved survival outcomes, emerging survivorship challenges including long-term treatment toxicities and second primary malignancies (SPMs) demand increased clinical attention. Notably, contemporary studies indicate that SPM incidence remains unaffected by the incorporation of rituximab into therapeutic regimens [3–6], emphasizing the critical need for optimized surveillance strategies and preventive measures in the current era of rituximab-dominated therapies.

Previous studies have showed that FL patients face a higher SPMs risk than the general population, with factors such as older age, male, chemotherapy, radiotherapy, radioimmunotherapy, autologous stem cell transplantation following high-dose chemotherapy, multiple treatments, and B symptoms linked to SPM development [2,5,7–10]. However, existing research presents conflicting conclusions regarding risk factor associations, compounded by methodological limitations: (1) most analysis predate widespread rituximab adoption, potentially obscuring modern risk profiles shaped by newer combination therapies (e.g., rituximab plus lenalidomide) [11]; (2) conventional Kaplan-Meier analysis overestimate SPM incidence by neglecting competing mortality risks; and (3) the clinical characteristics, survival outcomes, and

prognostic determinants for FL patients post-SPM diagnosis remain poorly characterized due to SPM rarity and insufficient longitudinal data.

To address these knowledge gaps, we conducted a population-based cohort study with three principal objectives: First, to identify clinical predictors of SPM development in FL patients treated during the rituximab era (post-2000). Second, to identify distinct prognostic determinants that differentiate OS from lymphoma-specific survival (LSS) between FL patients and the subgroup developing secondary malignancies following FL diagnosis. Third, to delineate cause-specific mortality patterns distinguishing these patient subgroups. Methodologically, we employ Fine-Gray competing risk models to provide more accurate SPM risk estimation compared to traditional survival analysis. Our findings aim to inform risk-adapted surveillance protocols and refine therapeutic decision-making in FL management.

In this population-based cohort study, we aimed to (1) identify clinical risk factors for SPMs in FL patients, (2) establish prognostic predictors for both overall survival (OS) and lymphoma-specific survival (LSS) comparing FL patients with versus without SPMs, and (3) delineate differential patterns of cause-specific mortality between these two clinically distinct populations.

## 2. Methods

### 2.1 Data source

The Surveillance, Epidemiology, and End Results (SEER) database, established by the National Cancer Institute (NCI), is a public database that systematically collects, curates, and maintains epidemiological and clinical data on cancer patients across the United States (available at: http://seer.cancer.gov/seerstat/).

This study extracted de-identified patient-level records from the SEER Research Data (17 Registries, November 2022 Submission, 2000−2020) using SEER*Stat 8.4.3. While the SEER-17 registries collectively cover approximately 26.5% of the U.S. population, their geographic distribution requires careful interpretation: The participating regions predominantly consist of selected states [California (excluding San Francisco/San Jose-Monterey/Los Angeles areas), Connecticut, Hawaii, Iowa, Kentucky, Louisiana, New Jersey, New Mexico, Utah] and metropolitan areas (San Francisco-Oakland, Los Angeles, Seattle-Puget Sound, Atlanta, Greater Georgia, Rural Georgia, San Jose-Monterey), with intentional over-sampling of specific populations (e.g., Alaska Natives). This deliberate sampling design may introduce geographic and demographic imbalances, particularly underrepresenting rural populations in non-SEER states. Three key limitations merit explicit acknowledgment. First, the absence of follicular lymphoma international prognostic index (FLIPI) scores prevents risk stratification, potentially masking survival differences among biological subgroups. Second, the absence of documented genetic biomarkers (e.g., BCL2 expression status, EZH2 mutation profiles, and chromosomal translocation patterns) hinders comprehensive assessment of genomic heterogeneity's prognostic implications. Third, while therapeutic modalities (chemotherapy, radiotherapy, and surgical resection) are recorded as binary variables, the lack of detailed treatment specifications including protocol variations, dose-intensity metrics, and temporal coordination of interventions may introduce confounding bias in comparative effectiveness research.

The data is public and patient information is anonymous, so our study does not require ethical approval or informed consent of patients. Our research follows the regulations published by the SEER database and the Declaration of Helsinki.

### 2.2 Patient selection

This retrospective cohort study analyzed clinical data and follow-up information from patients with follicular lymphoma from January 2000 and December 2020. Inclusion criteria were (1) first primary FL with histologically confirmed (ICD-O-3 histological codes 9690–9698); (2) age > 14 years; (3) complete and reliable follow-up data. The exclusion criteria included (1)did not have complete and reliable clinical characteristics; (2)diagnosis established post-mortem; (3)SPMs diagnosed within a six-month interval following the initial FL diagnosis; (4)SPMs of B-cell lineage, or documented relapse of FL; (5) survival time was 0. Notably, our sensitivity analysis included all SPM cases regardless of diagnostic interval(Tables S5–S15).The SPM definition followed National Cancer Institute guidelines as described by Morris et al [12], requiring

≥6-month interval between FL diagnosis and subsequent malignancy. As a population-based registry study utilizing the SEER-17 dataset (2000–2020), our analysis included all eligible follicular lymphoma cases meeting predefined criteria. This exhaustive sampling approach eliminates selection bias inherent to calculated sample sizes, ensuring maximal representation of the US population covered by SEER registries (covers approximately 26.5% of the U.S. population).

### 2.3 Analytical variables

Gender, age at diagnosis, race, ethnicity, Grade classification, Ann Arbor stage, B symptoms, treatment strategy, time from diagnosis to treatment, marital status, income level, residential area, primary site, year of diagnosis, incidence of SPMs, and cause of death were collected from the SEER database. The occurrence of SPMs, overall survival (OS), and lymphoma-specific survival (LSS) were the endpoints of our study. SPMs were defined as malignancies occurring more than 6 months after the diagnosis of primary FL. OS was defined as the time from diagnosis to death, regardless of the cause of death. LSS was defined as the time from diagnosis to death due to lymphoma. The time to the occurrence of a SPM was defined as the interval from the diagnosis of FL to the diagnosis of the SPM. Survival time was the duration from the initial diagnosis to death or to the last follow-up.

### 2.4 Statistical analysis

Baseline characteristics were analyzed using the Pearson $\chi 2$ test or Fisher's exact test for unordered categorical variables, ordinal categorical variables were compared using the rank-sum test. Continuous variables, such as age and time to development of SPMs, were described using median values, interquartile ranges, and the range of maximum and minimum values. Survival analysis was conducted using the Kaplan-Meier method, and the log-rank test and the Landmark test were used to compare the survival curves of the two groups.

We employed the Fine-Gray proportional subdistribution hazards model as the primary analytical framework to quantify associations between covariates and outcomes while accounting for the interdependence of competing events. Specifically, two distinct competing-risk regression analysis were conducted: (1) For SPMs, the primary event was defined as time from FL diagnosis to SPM detection, with all-cause mortality treated as the competing event; (2) For lymphoma-specific survival, the primary event was death from FL, with non-FL-related deaths constituting the competing event. Patients were censored at last follow-up if neither event occurred. Cumulative incidence functions were estimated using the Aalen-Johansen estimator to avoid overestimation biases inherent in Kaplan-Meier methodology. To address confounding, we implemented a rigorous covariate selection protocol: first, clinically plausible variables were pre-specified through literature review; second, univariate screening with an inclusive threshold (P ≤ 0.2) retained marginally significant factors; finally, multivariable models incorporated these predictors alongside established prognostic variables. Sensitivity analysis evaluated model robustness through subgroup stratification by SPM latency periods (≤6 vs. > 6 months post-diagnosis) and methodological cross-validation using conventional Cox proportional hazards models, confirming the stability of identified risk factors across analytical paradigms(Tables S5–S15).Cox hazards proportional analyses were performed to determine risk factors for all-cause mortality in FL patients and those with SPMs. Factors found to be statistically significant (P < 0.05), marginally significant (P ≤ 0.2), and those considered clinically relevant by the researchers were included in the multivariate analysis. P < 0.05 was indicative of statistical significance. All statistical analyses were performed using R software (version 4.4.1).

## 3.  Results

### 3.1 Epidemiology

Data from the SEER registry reveals that the incidence rate of FL in the American population decreased from 3.7 per 100,000 in 2000 to 2.8 per 100,000 in 2020, with a highest rate of 4.1 per 100,000 seen in year 2003−2004 and year 2007. The annual percent change (APC) from 2000 to 2020 was −1.6%.

In 2000, the incidence rate was 4.0 per 100,000 in males and 3.5 per 100,000 in females, peaking at 4.5 and 3.9 per 100,000 in year 2004 and year 2007, respectively. By 2020, these rates had gradually declined to 3.0 per 100,000 for males and 2.6 per 100,000 for females, with males generally showing a higher average incidence (3.8 per 100,000 versus 3.3 per 100,000).

Among different age groups, those aged ≥75 years, 65–74 years, 55–64 years, and 45–54 years saw peak incidence in year 2003–2004, reaching 12.6, 12.5, 8.7, and 4.6 per 100,000, respectively, but these rates decreased by 2020 to 8.7, 8.0, 5.2, and 3.1 per 100,000. The 15–44 age group had a relatively stable rate between 0.6 and 0.9 per 100,000 from 2000 to 2020. Individuals aged ≥65 consistently had the highest incidence among all age groups (12.5 per 100,000 versus 7.0 per 100,000 versus 3.9 per 100,000 versus 0.8 per 100,000) (S1 Fig, S1 Table).

By race and primary tumor site, FL incidence was higher in White individuals (4.0 per 100,000 versus 1.6 per 100,000 versus 1.8 per 100,000), and more patients with primary involvement of the lymph nodes (3.0 per 100,000 versus 0.5 per 100,000) (S1 Table).

### 3.2  Clinical characteristics

From 2000 to 2020, a total of 33,104 FL patients were included in the study. The median age at diagnosis was 62 years (IQR 52–72years), and the gender distribution was approximately equal (50.4% male, 49.6% female). Most patients were White (29,671/33,104, 89.6%), Grade 1–2 (17,409/33,104, 52.6%), and with advanced disease (13,750/33,104, 41.5%).

Among the cohort, 3,822 individuals (11.5%) with FL developed SPMs, with a median age at FL diagnosis of 70 years (IQR 62–77 years). In contrast, 29,282 patients (88.5%) remained free of SPMs, with a median age at FL diagnosis of 61 years (IQR 52–72 years). Compared with non-SPMs group, patients with SPMs were more likely to be male (55.3% vs. 49.8%, P < 0.0001), older (median age 70 years vs. 61 years), non-Hispanic (91.8% vs. 86.9%, P < 0.001), and White (91.6% vs. 89.4%, P < 0.001). SPMs group also included more non-urban residents (15.0% vs. 13.3%, P = 0.001), more advanced disease (47.5% vs. 40.8%, P < 0.001), a higher frequency of extranodal primary sites (15.0% vs. 13.5%, P = 0.006), and less B symptoms (5.6% vs. 10.1%, P < 0.001). A significantly higher proportion of patients in the SPMs group received treatment within one month of FL diagnosis (80.7% vs. 76.0%, P < 0.001). A greater percentage of individuals with SPMs underwent radiotherapy (21.4% vs. 18.9%, P < 0.001) or surgery (52.7% vs. 44.4%, P < 0.001) (Table 1).

Further analysis of treatment modalities and primary lymphoma sites showed that the SPMs group more frequently received combination therapy (38.8% vs. 31.6%, P < 0.001) than the non-SPMs group (S2 Table). The non-SPMs group had a higher proportion of patients with primary sites in intraperitoneal lymph nodes and multiple regions involved at diagnosis (8.7% vs. 7.2%, P = 0.002; 42.4% vs. 40.3%, P = 0.012), whereas the SPMs group more often had primary sites in the head, face, neck lymph nodes, and skin (13.1% vs. 10.5%, P < 0.001; 2.8% vs. 2.2%, P = 0.018) (S3 Table).

### 3.3  Risk of SPMs

Out of the total population, 3,822 patients (11.5%) had developed SPMs. The median time from first diagnose to SPMs was 61 months (IQR 29–105, range 6–245 months). The cumulative incidence rates for SPMs were 5.8% at 5 years, 9.4% at 10 years, and 11.1% at 15 years. The overall incidence rate of SPMs was 0.0016 per person-year. Among SPMs, lung tumors were the most common solid tumors, and myeloid/monocytic leukemias were the most common hematologic malignancies. (S4 Table).

Univariate competing-risk regression analysis revealed that radiotherapy (HR: 1.13, 95% CI: 1.04–1.22, P = 0.002) and primary skin involvement (HR: 1.28, 95% CI: 1.05–1.55, P = 0.014) significantly increased the risk of developing SPMs. Age between 40−60 years (HR: 2.66, 95% CI: 2.14–3.31, P < 0.001), age > 60 years (HR: 3.52, 95% CI: 2.83–4.37, P < 0.001), non-Hispanic ethnicity (HR: 1.41, 95% CI: 1.26–1.58, P < 0.001), and non-urban residency (HR: 1.10, 95% CI: 1.01–1.21, P = 0.027) were also identified as risk factors for the development of SPMs. Conversely, female (HR: 0.80, 95%

**Table 1. Characteristics of patients with follicular lymphoma at diagnosis: Surveillance, Epidemiology, and End Results (SEER) 2000-2020.**

| Characteristic(N,%) | All patients | Non-SPMs | SPMs | P value[a] | Time to SPMs, m | | |
|---|---|---|---|---|---|---|---|
| | | | | | Median[b] | IQR | Range |
| All patients | N = 33104 | 29282 (88.5%) | 3822 (11.5%) | – | 61 | 29-105 | 6-245 |
| **Sex** | | | | | | | |
| Male | 16683 (50.4%) | 14569 (49.8%) | 2114 (55.3%) | **<0.001** | 60 | 29-106 | 6-245 |
| Female | 16421 (49.6%) | 14713 (50.2%) | 1708 (44.7%) | | 62 | 29-105 | 6-240 |
| **Age at diagnosis** | | | | | | | |
| 15-39 | 1955 (5.9%) | 1872 (6.4%) | 83 (2.2%) | **<0.001** | 83 | 53-138 | 8-245 |
| 40-60 | 13423 (40.5%) | 11998 (41.0%) | 1425 (37.3%) | | 78 | 39-129 | 6-240 |
| >60 | 17726 (53.6%) | 15412 (52.6%) | 2314 (60.5%) | | 52.5 | 25-91 | 6-237 |
| **Race** | | | | | | | |
| White | 29671 (89.6%) | 26171 (89.4%) | 3500 (91.6%) | **<0.001** | 60 | 29-105 | 6-245 |
| Black | 1551(4.7%) | 1390 (4.7%) | 161 (4.2%) | | 68 | 34-117 | 6-208 |
| Others[c] | 1882 (5.7%) | 1721 (5.9%) | 161 (4.2%) | | 66 | 31.5-107 | 6-231 |
| **Ethnicity** | | | | | | | |
| Hispanics | 4161 (12.6%) | 3846 (13.1%) | 315 (8.2%) | **<0.001** | 59 | 29-114 | 6-237 |
| Non-Hispanics | 28943 (87.4%) | 25436 (86.9%) | 3507 (91.8%) | | 61 | 29-105 | 6-245 |
| **Marital status** | | | | | | | |
| Married | 21618(65.3%) | 18978(64.8%) | 2640(69.1%) | **<0.001** | 61 | 52-70 | 20-90 |
| Single | 4461(13.5%) | 4059(13.9%) | 402(10.5%) | | 54 | 45-64 | 15-90 |
| Others | 7025(21.2%) | 6245(21.3%) | 780(20.4%) | | 69 | 59-79 | 20-90 |
| **FL-subtype** | | | | | | | |
| Grade1–2 | 17409 (52.6%) | 15346 (52.4%) | 2063 (54.0%) | **0.007** | 61 | 29-105 | 6-245 |
| Grade3 | 6450 (19.5%) | 5779 (19.7%) | 671 (17.6%) | | 57 | 31-100 | 6-237 |
| Grade NOS | 9245 (27.9%) | 8157 (29.9%) | 1088 (28.5%) | | 63 | 29-103 | 6-237 |
| **Ann Arbor stage** | | | | | | | |
| I/ II | 10925 (33.0%) | 9342 (31.9%) | 1583 (41.4%) | **<0.001** | 63 | 32-112 | 6-239 |
| III/IV | 13750 (41.5%) | 11936 (40.8%) | 1814 (47.5%) | | 66 | 32-106 | 6-245 |
| Unknown | 8429 (25.5%) | 8004 (27.3%) | 425 (11.1%) | | 31 | 15-68 | 6-219 |
| **B symptom** | | | | | | | |
| None | 10836 (32.7%) | 10101 (34.5%) | 735 (19.2%) | **<0.001** | 36 | 18-59 | 6-125 |
| Any | 3176 (9.6%) | 2962 (10.1%) | 214 (5.6%) | | 36 | 18.75-68 | 6-127 |
| Unknown | 19092 (57.7%) | 16219 (55.4%) | 2873 (75.2%) | | 75 | 36-120 | 6-245 |
| **Radiotherapy** | 6361 (19.2%) | 5543 (18.9%) | 818 (21.4%) | **<0.001** | 68.5 | 34-119 | 6-239 |
| **Chemotherapy** | 22138 (66.9%) | 19574 (66.8%) | 2564 (67.0%) | 0.451 | 60 | 29-105 | 6-245 |
| **Surgery** | 15031 (45.4%) | 13015 (44.4%) | 2016 (52.7%) | **<0.001** | 67 | 32-113 | 6-237 |
| **Diagnosis-to-treatment-time** | | | | | | | |
| ≤1month | 25345 (76.6%) | 22261 (76.0%) | 3084(80.7%) | **<0.001** | 64 | 31-109 | 6-240 |
| > 1month | 7759 (23.4%) | 7021 (24.0%) | 738 (19.3%) | | 51 | 24.75-90 | 6-245 |
| **Income** | | | | | | | |
| <$65,000 | 10430 (31.5%) | 9167 (31.3%) | 1263 (33.0%) | 0.065 | 59 | 28-100 | 6-245 |
| $65,000 - $74,999 | 8492 (25.6%) | 7530 (25.7%) | 962 (25.2%) | | 60 | 30-101 | 6-231 |
| ≥$75,000 | 14182 (42.8%) | 12585 (43.0%) | 1597 (41.8%) | | 63 | 30-113 | 6-240 |
| **Rural-Ubran** | | | | | | | |
| Metropolitan areas | 28636 (86.5%) | 25389 (86.7%) | 3247 (85.0%) | **0.001** | 61 | 29-105 | 6-245 |
| Nonmetropolitan counties | 4468 (13.5%) | 3893 (13.3%) | 575 (15.0%) | | 60 | 29-107 | 6-240 |
| **Site** | | | | | | | |

*(Continued)*

**Table 1.** (Continued)

| Characteristic(N,%) | All patients | Non-SPMs | SPMs | P value[a] | Time to SPMs, m | | |
|---|---|---|---|---|---|---|---|
| | | | | | Median[b] | IQR | Range |
| NHL – Extranodal | 4527 (13.7%) | 3957 (13.5%) | 570 (15.0%) | **0.006** | 61 | 31-109 | 6-239 |
| NHL – Nodal | 28577 (86.3%) | 25325 (86.5%) | 3252 (85.0%) | | 61 | 29-105 | 6-245 |
| **Year of diagnosis** | | | | | | | |
| 2000-2004 | 8299 (25.1%) | 6864 (23.4%) | 1435 (37.5%) | **<0.001** | 86 | 43-139 | 6-245 |
| 2005-2009 | 8727 (26.4%) | 7444 (25.4%) | 1283 (33.6%) | | 70 | 34-109 | 6-182 |
| 2010-2014 | 7372 (22.3%) | 6607 (22.6%) | 765 (20.1%) | | 44 | 23.5-74 | 6-131 |
| 2015-2019 | 7491 (22.6%) | 7159 (22.4%) | 332 (8.6%) | | 22.5 | 14-36 | 6-68 |
| 2020 | 1215 (3.7%) | 1208 (4.1%) | 7 (0.2%) | | 7 | 6-8 | 6-10 |

SPMs, second primary malignances; IQR, interquartile range; Grade NOS, Not Otherwise Specified; NHL, non-Hodgkin's lymphoma.

[a] χ2 test was used for comparison. Significant values (P<0.05) are highlighted in bold.

[b] Kruskall-Wallis test used for calculations and comparisons.

[c] Others for race represented American Indian/AK Native, Asian/Pacific Islander.

[d] Others for marital status represented divorced, separated, unmarried or domestic partner, widowed.

CI: 0.75–0.85, P<0.001), primary lymph node involvement (HR: 0.90, 95% CI: 0.82–0.98, P=0.019), and Grade 3 (HR: 0.89, 95% CI: 0.81–0.97, P=0.007) were associated with a lower risk of developing SPMs (Fig 1).

Multivariate competing-risk regression found that patients aged 40−60 years (HR: 2.60, 95% CI: 2.08–3.24, P<0.001),>60 years (HR: 3.51, 95% CI: 2.82–4.38, P<0.001), and non-Hispanic ethnicity (HR: 1.28, 95% CI: 1.14–1.44, P<0.001), were associated with an increased risk of SPMs. In contrast, female patients (HR: 0.79, 95% CI: 0.74–0.84, P<0.001), single status (HR: 0.90, 95% CI: 0.81–1.00, P=0.049), initial treatment without radiotherapy (HR: 0.90, 95% CI: 0.82–0.97, P=0.009), Grade 3 (HR: 0.89, 95% CI: 0.82–0.98, P=0.012) and those diagnosis between 2015−2019(HR: 0.75, 95% CI: 0.60–0.93, P=0.011)had a reduced risk of developing SPMs (Fig 2).

### 3.4 Survival analysis

In our cohort, 11,562 FL patients (34.9%) died, with a crude mortality rate of 0.045 per person per year. Patients who remained event-free by the study termination date (December 31, 2020) or were lost to follow-up underwent right-censoring at their last confirmed survival date. For survival analysis, distinct endpoints were defined: overall survival (OS) considered all-cause mortality as the event of interest, while lymphoma-specific survival (LSS) specifically captured deaths attributable to follicular lymphoma. The Kaplan-Meier analysis estimated the 1-, 3-, 5-, and 10-year survival probabilities as follows: OS rates were 93.2% (95% CI: 93.0–93.5), 85.6% (95% CI: 85.2–86.0), 79.0% (95% CI: 78.5–79.5), and 64.3% (95% CI: 63.7–64.9), respectively; while LSS rates demonstrated superior outcomes at 94.8% (95% CI: 94.6–95.1), 89.9% (95% CI: 89.6–90.3), 86.4% (95% CI: 86.0–86.8), and 79.9% (95% CI: 79.4–80.4) for the corresponding time intervals. The median OS was 192 months (95% CI: 189−195), whereas the median LSS remained unreached at the time of final analysis. In the SPMs group, 2,053 patients died (2,053/3,822, 53.7%), compared to 9,509 deaths (9,509/29,282, 32.5%) in the non-SPMs group, indicating a significantly higher mortality rate in the SPMs group (P<0.001). The 5-year OS rates for the SPMs and non-SPMs groups were 83.5% (95% CI: 82.3–84.7) and 78.4%(95% CI: 77.9–78.9), with median OS of 150 months(95% CI: 145−158) and 210 months(95% CI: 205−217), respectively (P<0.001). The 5-year LSS rates were 93.3%(95% CI: 92.5–94.2) for the SPMs group and 85.4% (95% CI: 85.0–85.8)for the non-SPMs group. The median LSS was not attained in either cohort during the study period. Mortality risk in the SPMs group increased significantly 91 months after FL diagnosis, and lymphoma-specific death risk increased 175 months after diagnosis (Fig 3).

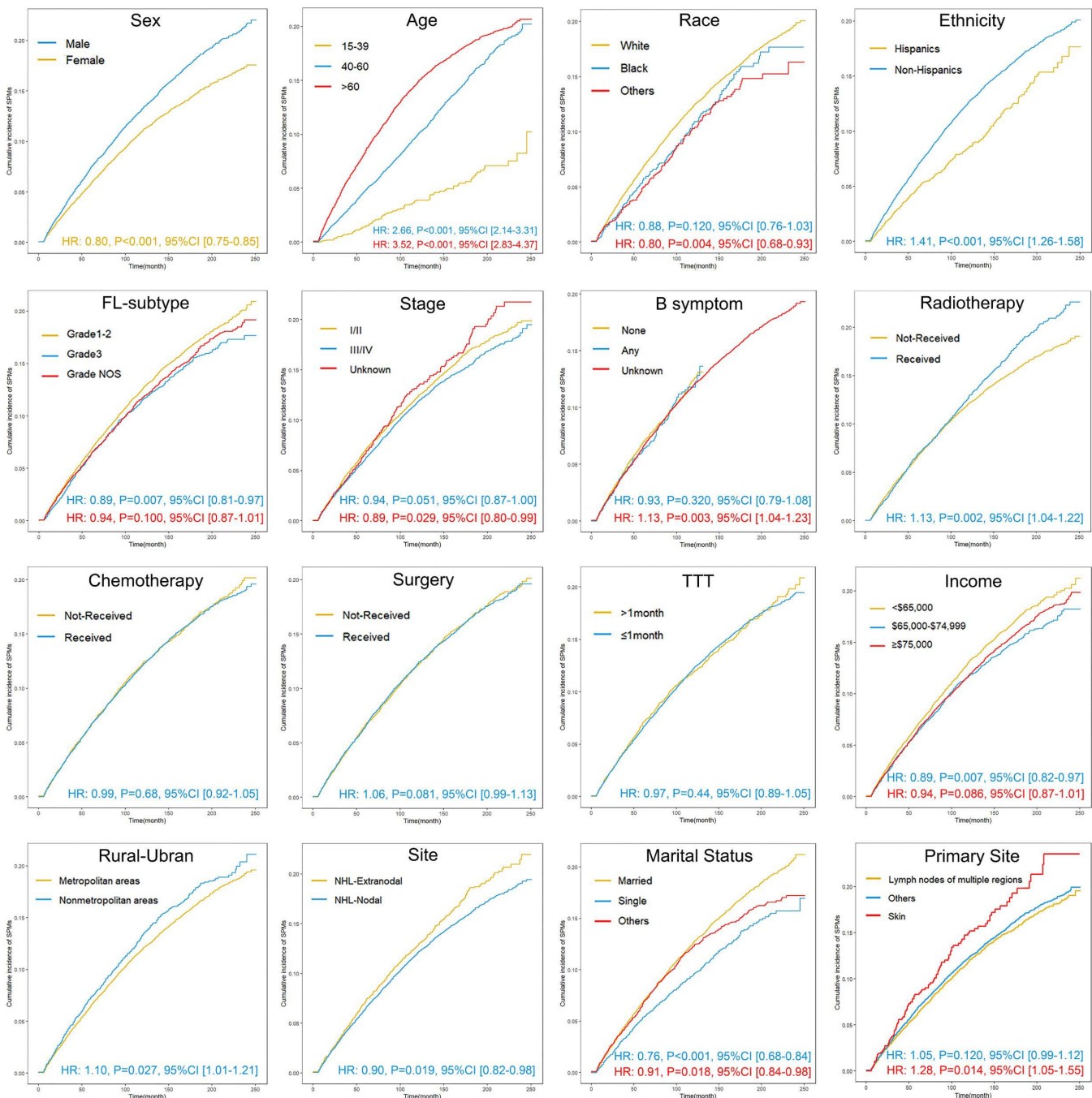

**Fig 1. Cumulative incidence of second primary malignancies in patients with follicular lymphoma diagnosed between 2000 and 2020.** TTT, time to treatment; NHL, non-Hodgkin's lymphoma; Grade NOS, Not Otherwise Specified. Others of the race: American Indian/AK Native, Asian/Pacific Islander. Others of marital status: divorced, separated, unmarried or domestic partner, widowed.

## Forestplot

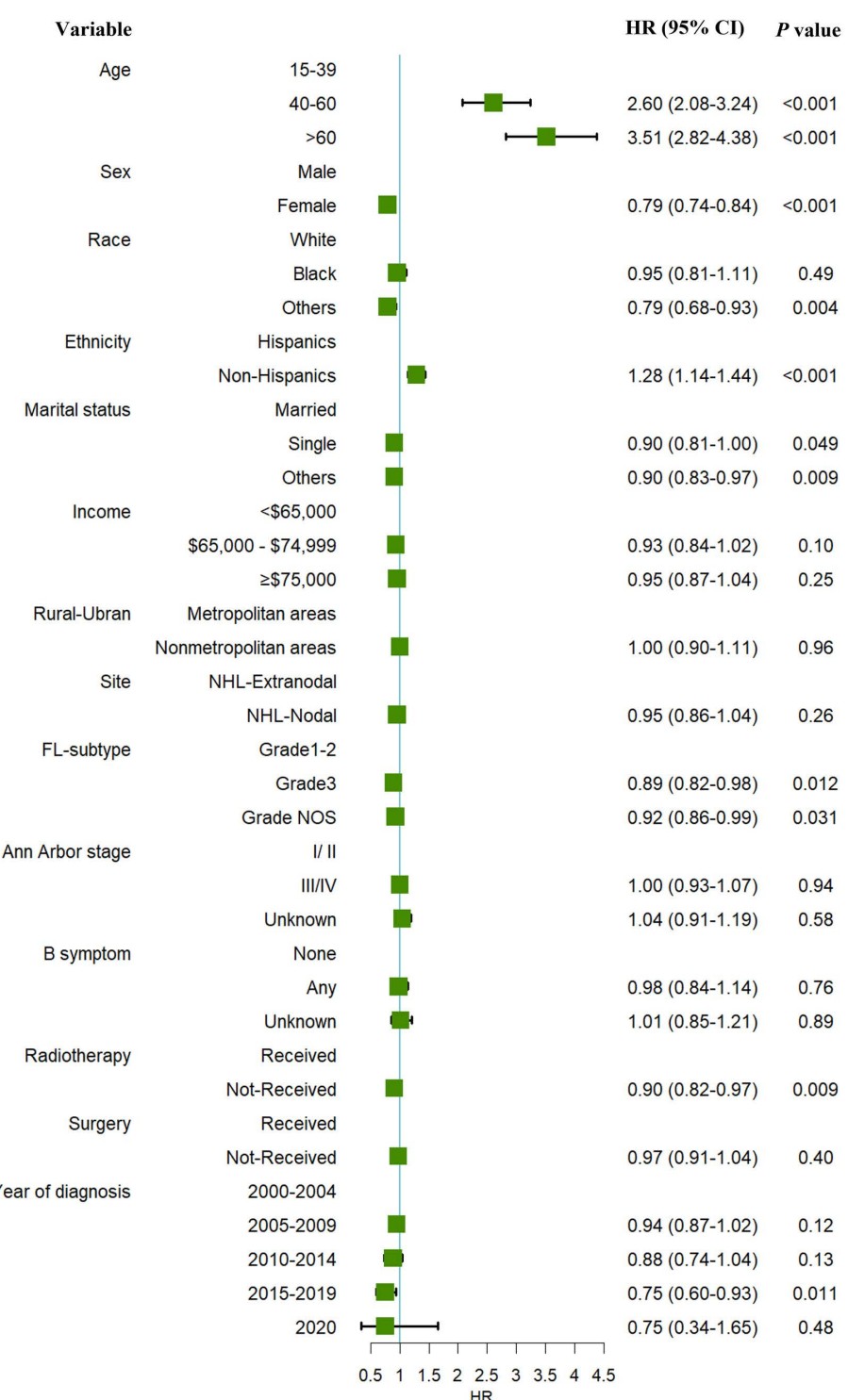

| Variable | | HR (95% CI) | *P* value |
|---|---|---|---|
| Age | 15-39 | | |
| | 40-60 | 2.60 (2.08-3.24) | <0.001 |
| | >60 | 3.51 (2.82-4.38) | <0.001 |
| Sex | Male | | |
| | Female | 0.79 (0.74-0.84) | <0.001 |
| Race | White | | |
| | Black | 0.95 (0.81-1.11) | 0.49 |
| | Others | 0.79 (0.68-0.93) | 0.004 |
| Ethnicity | Hispanics | | |
| | Non-Hispanics | 1.28 (1.14-1.44) | <0.001 |
| Marital status | Married | | |
| | Single | 0.90 (0.81-1.00) | 0.049 |
| | Others | 0.90 (0.83-0.97) | 0.009 |
| Income | <$65,000 | | |
| | $65,000 - $74,999 | 0.93 (0.84-1.02) | 0.10 |
| | ≥$75,000 | 0.95 (0.87-1.04) | 0.25 |
| Rural-Ubran | Metropolitan areas | | |
| | Nonmetropolitan areas | 1.00 (0.90-1.11) | 0.96 |
| Site | NHL-Extranodal | | |
| | NHL-Nodal | 0.95 (0.86-1.04) | 0.26 |
| FL-subtype | Grade1-2 | | |
| | Grade3 | 0.89 (0.82-0.98) | 0.012 |
| | Grade NOS | 0.92 (0.86-0.99) | 0.031 |
| Ann Arbor stage | I/ II | | |
| | III/IV | 1.00 (0.93-1.07) | 0.94 |
| | Unknown | 1.04 (0.91-1.19) | 0.58 |
| B symptom | None | | |
| | Any | 0.98 (0.84-1.14) | 0.76 |
| | Unknown | 1.01 (0.85-1.21) | 0.89 |
| Radiotherapy | Received | | |
| | Not-Received | 0.90 (0.82-0.97) | 0.009 |
| Surgery | Received | | |
| | Not-Received | 0.97 (0.91-1.04) | 0.40 |
| Year of diagnosis | 2000-2004 | | |
| | 2005-2009 | 0.94 (0.87-1.02) | 0.12 |
| | 2010-2014 | 0.88 (0.74-1.04) | 0.13 |
| | 2015-2019 | 0.75 (0.60-0.93) | 0.011 |
| | 2020 | 0.75 (0.34-1.65) | 0.48 |

**Fig 2. Forest plot of multivariate competing risks regression analysis for risks of second primary malignancies in follicular lymphoma patients.** NHL, non-Hodgkin's lymphoma; Grade NOS, Not Otherwise Specified. Others of the race: American Indian/AK Native, Asian/Pacific Islander. Others of marital status: divorced, separated, unmarried or domestic partner, widowed.

 

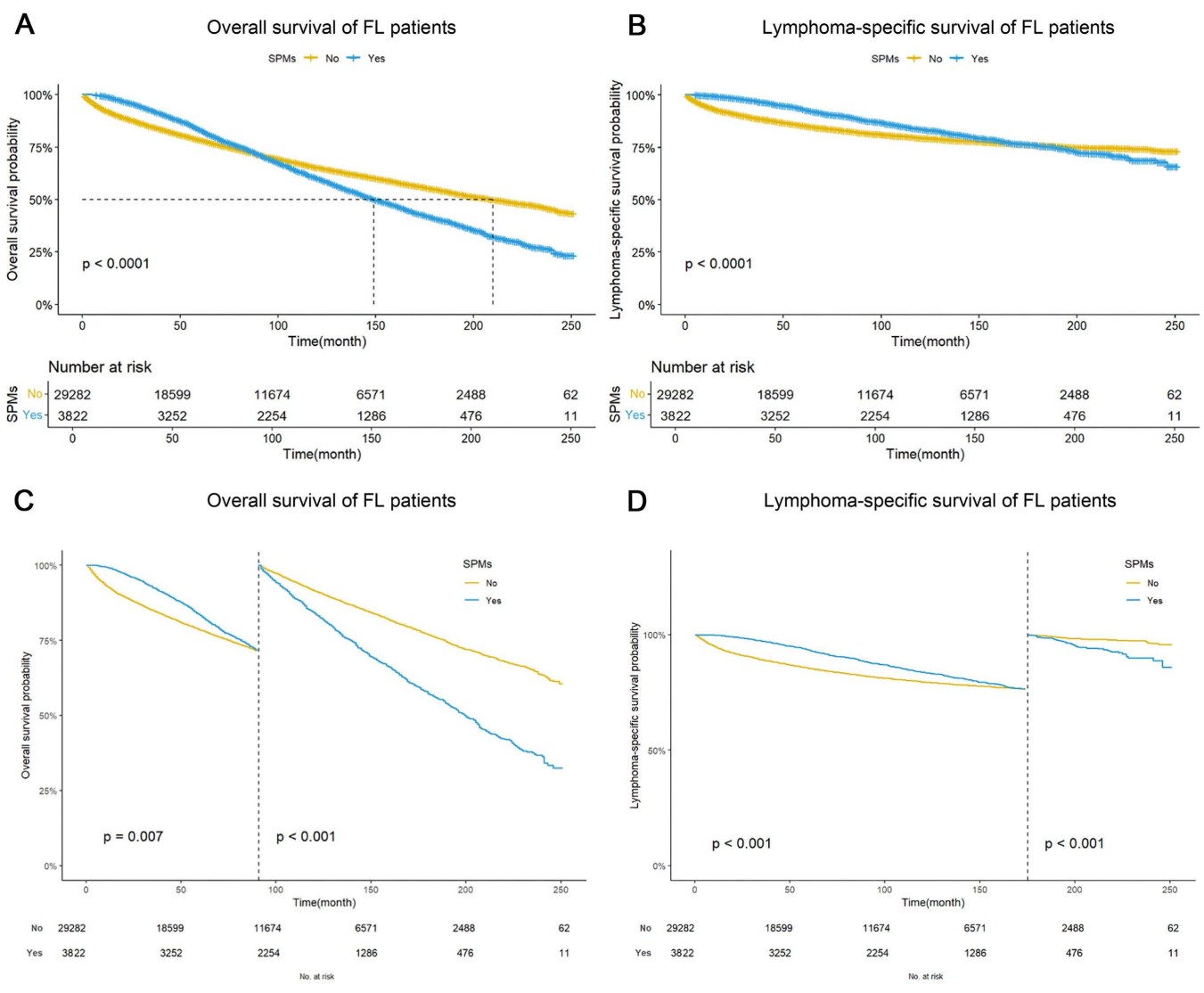

**Fig 3. Survival analysis of follicular lymphoma patients based on secondary primary malignancy status.** (A) Overall survival (OS) stratified by occurrence of secondary primary malignancies (SPMs);(B) Lymphoma-specific survival (LSS) stratified by occurrence of SPMs; (C) Log-rank test comparison of OS between patients with and without SPMs; (D) Log-rank test comparison of LSS between patients with and without SPMs.

Further analysis of causes of death among FL patients revealed that 5,732 patients (49.6%) died from lymphoma, 5,721 patients (49.5%) died from non-malignant diseases, and 109 patients (0.9%) died from other cancers. The top five causes of death were lymphoma, heart disease, Chronic Obstructive Pulmonary Disease (COPD), cerebrovascular disease, and lung infection. Patients in the SPMs group had a significantly higher rate of death from other cancers compared to the non-SPMs group (5.3% vs. 0.0%, P < 0.001), making other cancers the fifth leading cause of death in the SPM group. Among non-malignant causes of death, the SPMs group had a lower proportion of deaths due to heart disease and COPD compared to the non-SPMs group (7.7% vs. 16.6%, P < 0.001; 2.9% vs. 3.2%, P < 0.001) (Table 2).

**Table 2. Causes of death in patients with follicular lymphoma.**

| Death cause (N, %) | All deaths (N = 11,562) | Non-SPMs (N = 9,509) | SPMs (N = 2,053) | ᵃp value |
|---|---|---|---|---|
| Lymphoma | 5,732 (49.6%) | 5,060(53.2%) | 672(32.7%) | 0.726 |
| Another Malignancy | 109 (0.9%) | 1(0.0%) | 108(5.3%) | **<0.001** |
| Non-cancer causes | 5,721 (49.5%) | 4,448(46.8%) | 1,273(62.0%) | **<0.001** |
| Diseases of Heart | 1,737 (15.0%) | 1,578(16.6%) | 159(7.7%) | **0.001** |
| COPD | 363 (3.1%) | 304(3.2%) | 59(2.9%) | **0.003** |
| Cerebrovascular Diseases | 301 (2.6%) | 275(2.9%) | 26(1.3%) | 0.106 |
| Pneumonia and Influenza | 245 (2.1%) | 219(2.3%) | 26(1.3%) | 0.627 |
| Alzheimer's Disease | 220 (1.9%) | 201(2.1%) | 19(0.9%) | 0.168 |
| Septicemia | 117 (1.0%) | 99(1.0%) | 18(0.9%) | 0.201 |
| Other Non-cancer causes | 2,738(23.7%) | 1,772(18.6%) | 966(47.1%) | **<0.001** |

ᵃ χ2 test was used for comparison. Significant values (P < 0.05) are highlighted in bold.

COPD: chronic obstructive pulmonary disease.

### 3.5 Prognostic factors among FL patients

A multivariate analysis of FL patients found that age ≥ 40 years, Black race, single status, non-urban residency, primary lymphoma site within lymph nodes, Grade 3, advanced Ann Arbor stage, and B symptoms were independent risk factors for both OS and LSS. Additionally, FL patients who received chemotherapy as their initial treatment had a worse LSS. Female, higher income, diagnosis after 2005, diagnosis-to-treatment time (DTT) over one month, and radiotherapy or surgery were linked to reduced risks of both OS and LSS. The development of SPMs was associated with a decreased risk of LSS in FL patients (Table 3).

### 3.6 Prognostic factors among FL patients developing SPMs

Multivariate analysis revealed that FL patients over 60 years, those who were single, and those diagnosed after 2010 faced a higher risk of OS after developing SPMs. Additionally, initial radiotherapy faced a lower risk of both OS and LSS among patients who developed SPMs (Table 4).

## 4. Discussion

Follicular lymphoma (FL) is a typically indolent lymphoma originating from germinal center B cells. Advances in treatment have significantly improved FL patient survival [13]. In the era of rituximab therapy, patients diagnosed with FL generally have a promising prognosis. Recent data from French and U.S. cohorts show 10-year OS rates of FL patients were 79.8% and 76.6%, respectively [14], although patients may still experience late effects following cancer treatment, impacting their quality of life.

　Previous studies have reported that the APC for FL in Taiwan from 2002 to 2019 was 3.49%, in Japan from 2001 to 2008 was 12.66%, and from 2014 to 2019 was 4.95%. In South Korea, the change was 5.72% from 2001 to 2012, and 7.93% from 2015 to 2022 [15]. In contrast, the incidence rate among the Whites decreased annually by −2.1% from 2008 to 2017, dropping from 3.42 cases in 2008 to 2.74 cases per 100,000 person-years in 2017, a decrease of 20% [16]. Between 2000 and 2014 within the Polish population, no significant trend was noted in the standardized overall incidence rates of FL [17]. Consistent with the findings of the above studies, our research shows an APC of −1.6% for FL in the U.S. population from 2000 to 2020. This suggests that greater attention to FL in Asian countries is needed, along with etiological studies that focus especially on modifiable risk factors, to understand and address the factors driving the rising incidence of FL group.

**Table 3. Multivariate analysis of prognostic factors in patients with follicular lymphoma after initial diagnosis.**

| Characteristics | Overall survival[a] | | Lymphoma-specific survival[b] | |
|---|---|---|---|---|
| | HR (95% CI) | P | HR (95% CI) | P |
| **Sex** | | | | |
| Male | 1 | – | 1 | – |
| Female | 0.73 (0.71-0.76) | **<0.001** | 0.76 (0.72-0.80) | **<0.001** |
| **Age** | | | | |
| 15-39 | 1 | – | 1 | – |
| 40-60 | 1.96 (1.72-2.25) | **<0.001** | 1.67 (1.41-1.97) | **<0.001** |
| >60 | 6.92 (6.06-7.90) | **<0.001** | 4.01 (3.40-4.73) | **<0.001** |
| **Race** | | | | |
| White | 1 | – | 1 | – |
| Black | 1.13 (1.04-1.23) | **0.006** | 1.23 (1.09-1.38) | **<0.001** |
| Others | 0.93 (0.85-1.02) | 0.12 | 1.08 (0.95-1.22) | 0.24 |
| **Ethnicity** | | | | |
| Hispanics | 1 | – | – | – |
| Non-Hispanics | 1.00 (0.94-1.07) | 0.977 | 0.90(0.83-0.98) | **0.018** |
| **Marital status** | | | | |
| Married | 1 | – | 1 | – |
| Single | 1.42 (1.33-1.50) | **<0.001** | 1.29 (1.19-1.41) | **<0.001** |
| other | 1.73 (1.66-1.81) | **<0.001** | 1.50 (1.41-1.59) | **<0.001** |
| **Income** | | | | |
| <$65,000 | 1 | – | 1 | – |
| $65,000 - $74,999 | 0.97 (0.92-1.02) | 0.204 | 0.99 (0.92-1.06) | 0.75 |
| ≥$75,000 | 0.83 (0.79-0.87) | **<0.001** | 0.86 (0.81-0.93) | **<0.001** |
| **Rural-Ubran** | | | | |
| metropolitan areas | 1 | – | 1 | – |
| Nonmetropolitan counties | 1.07 (1.01-1.13) | **0.028** | 1.11 (1.02-1.21) | **0.014** |
| **Year of diagnosis** | | | | |
| 2000-2004 | 1 | – | 1 | – |
| 2005-2009 | 0.81 (0.77-0.84) | **<0.001** | 0.66 (0.62-0.71) | **<0.001** |
| 2010-2014 | 0.72 (0.66-0.80) | **<0.001** | 0.58 (0.50-0.66) | **<0.001** |
| 2015-2019 | 0.65 (0.58-0.74) | **<0.001** | 0.55 (0.46-0.65) | **<0.001** |
| 2020 | 0.67 (0.47-0.96) | **0.029** | 0.53 (0.33-0.85) | **0.008** |
| **Site** | | | | |
| NHL – Extranodal | 1 | – | 1 | – |
| NHL – Nodal | 1.11 (1.05-1.18) | **<0.001** | 1.26 (1.15-1.38) | **<0.001** |
| **FL-subtype** | | | | |
| Grade1–2 | 1 | – | 1 | – |
| Grade3 | 1.08 (1.03-1.14) | **0.002** | 1.09 (1.01-1.17) | **0.02** |
| Grade NOS | 1.19 (1.14-1.24) | **<0.001** | 1.25 (1.18-1.33) | **<0.001** |
| **Ann Arbor stage** | | | | |
| I/ II | 1 | – | 1 | – |
| III/IV | 1.28 (1.22-1.33) | **<0.001** | 1.57 (1.47-1.68) | **<0.001** |
| Unknown | 1.10 (1.02-1.19) | **0.02** | 1.08 (0.96-1.22) | 0.17 |
| **B symptom** | | | | |
| None | 1 | – | 1 | – |
| Any | 1.41 (1.29-1.53) | **<0.001** | 1.56 (1.39-1.76) | **<0.001** |

*(Continued)*

**Table 3.** (Continued)

| Characteristics | Overall survival[a] | | Lymphoma-specific survival[b] | |
|---|---|---|---|---|
| | HR (95% CI) | P | HR (95% CI) | P |
| Unknown | 1.27 (1.14-1.40) | **<0.001** | 1.51 (1.31-1.74) | **<0.001** |
| **Diagnosis-to-treatment time** | | | | |
| ≤1month | 1 | – | 1 | – |
| >1month | 0.81 (0.77-0.85) | **<0.001** | 0.71 (0.66-0.77) | **<0.001** |
| **Radiotherapy** | | | | |
| Received | 1 | – | 1 | – |
| Not received | 1.17(1.11-1.23) | **<0.001** | 1.15(1.06-1.25) | **<0.001** |
| **Chemotherapy** | | | | |
| Received | 1 | – | 1 | – |
| Not received | 0.98 (0.94-1.03) | 0.45 | 0.81(0.75-0.87) | **<0.001** |
| **Surgery** | | | | |
| Received | 1 | – | 1 | – |
| Not received | 1.19 (1.14-1.25) | **<0.001** | 1.23(1.15-1.31) | **<0.001** |
| **SPMs** | | | | |
| No | 1 | – | 1 | – |
| Yes | 1.08 (1.03-1.13) | **<0.001** | 0.66(0.61-0.71) | **<0.001** |

FL, follicular lymphoma; NHL, non-Hodgkin's lymphoma; SPMs, second primary malignancies.

[a] Cox hazards proportional analysis of prognostic factors for patients with follicular lymphoma. Significant values (P<0.05) are highlighted in bold.

[b] Competing-risk regression analysis of prognostic factors for patients died from follicular lymphoma. Deaths from causes other than follicular lymphoma were considered competing events. Significant values (P<0.05) are highlighted in bold.

Among 33,104 FL patients in our study, the median age at diagnosis was 62 years, with a balanced male-to-female ratio, matching typical FL incidence patterns [1,18]. Consistent with prior research, male and older patients had a higher incidence of SPMs [19]. Notably, a higher proportion of patients with skin as the primary site developed SPMs compared to those without such a history. While the existing literature on this is limited, we hypothesize that this may be related to the immunosuppressive therapies, such as radiation therapy (a common treatment for primary cutaneous follicular lymphoma) and chemotherapy [20]. In our study, the SPM subtypes significantly elevated in FL patients are those thought to be linked to immune dysfunction and the use of chemotherapeutic agents, including acute myeloid leukemia (AML) and malignancies of the lung and bronchus, skin [21].

Giri et al. [22] analyzed SEER data (1992–2011; N = 15,517) excluding ICD-O-3 9690/3 (FL NOS) cases due to insufficient diagnostic specificity, applying Fine-Gray competing risk models. Their analysis identified age > 65 years, male sex, and prior radiotherapy(RT) as independent SPM predictors, while no significant associations observed for diagnosis year (1992–1999 vs 2000–2011), FL histological grade, or Black/White racial disparities. Our findings corroborate Giri et al.'s established SPM risk factors (age>65 years, male sex, RT) while identifying novel predictors (non-Hispanic ethnicity) and protective factors [unmarried status, Grade 3 histology, recent diagnosis (2015-2019)] through competing risk analysis. The underlying mechanisms contributing to the reduced incidence of SPMs in grade 3 FL remain underexplored in current literature, highlighting a critical knowledge gap in FL pathogenesis. A pivotal competing risk analysis demonstrated significantly higher cumulative histological transformation(HT) rates in FL3a patients compared to their FL1-2 counterparts, with 10% (5-year) and 13% (10-year) versus 2.1% and 2.9%, respectively (P < 0.001) [23]. This striking divergence suggests potential biological competition between transformation events and secondary carcinogenesis. Microenvironmental remodeling is a well-known oncogenic and procarcinogenic factor in FL. Through transcriptome analysis, Zhang [24] group explored the relationship between FL pathological grade and microenvironmental composition. From FL1-2

**Table 4. Multivariate analysis of prognostic factors in patients with follicular lymphoma after SPMs (N = 3822).**

| Characteristics | Overall survival[a] | | Lymphoma-specific survival[b] | |
|---|---|---|---|---|
| | HR (95% CI) | P | HR (95% CI) | P |
| **Age** | | | | |
| 15-39 | 1 | – | – | – |
| 40-60 | 1.24 (0.87-1.77) | 0.238 | – | – |
| >60 | 2.72 (1.91-3.88) | **<0.001** | – | – |
| **Race** | | | | |
| White | | | | |
| Black | 0.98(0.78-1.22) | 0.84 | | |
| Others | 0.86(0.68-1.08) | 0.19 | | |
| **Marital status** | | | | |
| Married | 1 | – | 1 | – |
| Single | 1.26 (1.09-1.47) | **0.002** | 1.17 (0.91-1.50) | 0.22 |
| other | 1.38 (1.25-1.53) | **<0.001** | 1.26 (1.05-1.51) | **0.013** |
| **Income** | | | | |
| <$65,000 | 1 | – | – | – |
| $65,000 - $74,999 | 1.02 (0.90-1.15) | 0.811 | 0.86(0.71-1.06) | 0.16 |
| ≥$75,000 | 0.86 (0.76-0.96) | **0.008** | 0.87(0.73-1.04) | 0.13 |
| **Rural-Ubran** | | | | |
| metropolitan areas | 1 | – | – | – |
| Nonmetropolitan counties | 1.09 (0.96-1.25) | 0.193 | – | – |
| **Year of diagnosis** | | | | |
| 2000-2004 | 1 | – | – | – |
| 2005-2009 | 1.06 (0.95-1.18) | 0.288 | 0.85(0.71-1.00) | 0.056 |
| 2010-2014 | 1.74(1.34-2.27) | **<0.001** | 0.88(0.70-1.10) | 0.27 |
| 2015-2020 | 2.73(1.93-3.85) | **<0.001** | 0.72(0.45-1.15) | 0.17 |
| **FL-subtype** | | | | |
| Grade1–2 | 1 | – | | |
| Grade3 | 1.00(0.89-1.13) | 0.996 | | |
| Grade NOS | 1.05(0.95-1.16) | 0.388 | | |
| **Ann Arbor stage** | | | | |
| I/ II | 1 | – | 1 | – |
| III/IV | 1.10 (0.99-1.21) | 0.076 | 1.14 (0.96-1.35) | 0.14 |
| Unknown | 1.09 (0.91-1.31) | 0.35 | 0.98 (0.70-1.36) | 0.88 |
| **B symptom** | | | | |
| None | 1 | – | – | – |
| Any | 1.01 (0.79-1.29) | 0.930 | – | – |
| Unknown | 1.16 (0.901-1.49) | 0.139 | – | – |
| **Radiotherapy** | | | | |
| Received | 1 | – | 1 | – |
| Not received | 1.15(1.06-1.33) | **0.019** | 1.30 (1.05-1.61) | **0.018** |
| **Chemotherapy** | | | | |
| Received | 1 | – | 1 | – |
| Not received | 0.97(0.7-1.08) | 0.60 | 0.90(0.75-1.09) | 0.17 |

*(Continued)*

**Table 4.** (Continued)

| Characteristics | Overall survival[a] | | Lymphoma-specific survival[b] | |
|---|---|---|---|---|
| | HR (95% CI) | P | HR (95% CI) | P |
| **Surgery** | | | | |
| Received | 1 | – | 1 | – |
| Not received | 1.10 (0.99-1.21) | 0.067 | 1.09(0.92-1.30) | 0.31 |

FL, follicular lymphoma.

[a] Cox hazards proportional analysis of prognostic factors for patients with follicular lymphoma. Significant values (P<0.05) are highlighted in bold.

[b] Competing-risk regression analysis of prognostic factors for patients died from follicular lymphoma. Deaths from causes other than follicular lymphoma were considered competing events. Significant values (P<0.05) are highlighted in bold.

to FL3B, the cell cycle pathways become progressively more active, metabolic activity gradually increases, and immune pathway activity gradually decreases. These dynamic molecular shifts may create an evolutionary landscape favoring rapid clonal expansion over secondary carcinogenic processes. However, this hypothesis requires validation through multi-omics investigations characterizing the distinct genomic landscapes, tumor microenvironment interactions, and clonal evolutionary patterns distinguishing HT-prone from SPM-susceptible FL subtypes. The protective effect of recent diagnosis (2015-2019) may be attributed to advancements in FL management. First, the widespread adoption of observation without immediate therapy for asymptomatic advanced-stage FL effectively delays or avoids alkylating agents (e.g., cyclophosphamide) and radiation, both recognized as independent risk factors for therapy-related myeloid neoplasms and solid tumors. Second, for early-stage FL requiring intervention, precision radiotherapy techniques have replaced extended-field irradiation. Crucially, the therapeutic landscape has shifted toward targeted agents with lower mutagenic potential: CD20-directed monoclonal antibodies, PI3Kδ inhibitors, and CAR-T cells targeting CD19/20 now dominate relapse/refractory settings, displacing traditional regimens like CHOP that confer higher cumulative DNA damage [2]. The association between unmarried status and reduced SPM risk is intriguing. As cancer diagnosis and treatment often create prolonged psychosocial burdens for both patients and spouses, remaining single potentially circumvents these marital stressors that studies suggest marital quality is associated with health outcomes and mortality risk [25]. Notably, our findings contrast with Nie et al.'s [26] logistic regression analysis (2003–2014 cohort) that applied full covariate adjustment and demonstrated no radiotherapy-second primary cancer (SPC) association (OR = 0.92, 95%CI: 0.53–1.58) in FL patients with ≥1-year latency between malignancies, potentially reflecting methodological differences in outcome definition (SPC latency ≥1year) and analytical approaches (competing risks vs conventional regression). Our findings highlight critical clinical considerations for managing FL patients at risk of SPMs. Clinicians should prioritize SPM surveillance in high-risk subgroups, including males≥40years, non-Hispanic individuals, and those treated with radiotherapy (RT). RT is a powerful treatment that can effectively destroy cancer but also poses risks to healthy tissue. When radiation doses exceed the repair capacity of surrounding normal tissues, irreversible damage may occur [27]. RT can trigger immunosuppressive responses, including the polarization of macrophages towards an M2 phenotype [28], the shift of neutrophils to an N2 phenotype [29], and the accumulation of myeloid-derived suppressor cells [30]. These immune cells typically exert their immunosuppressive influence through the upregulation of PD-L1 and through various other mechanisms [31]. To mitigate risks, we propose a two-pronged approach: (1) Enhanced Surveillance: Implement annual low-dose CT screening for high-risk patients to improve early lung cancer detection and survival [32]. (2) Treatment Optimization: Employ advanced RT techniques (e.g., involved-site RT) to minimize radiation exposure to healthy tissues. Treatment decisions should weigh histologic grade (Grade 3 cases showed reduced SPM risk) and marital status (unmarried patients had lower risk). For recently diagnosed cases (2015–2019 cohort with lower SPM risk), analyze modern treatment protocols as potential models for risk reduction. Future research should clarify how immunosuppressive therapies interact with FL biology to drive SPMs, which could guide targeted interventions.

The survival analysis found that FL patients with SPMs did not show a decrease in survival within the first 91 months compared to those without SPMs, with LSS increasing only after 175 months. This could be attributed to the early identification of high-risk patients in lymphoma, targeted therapies, cellular treatments, and other interventions have demonstrated promising efficacy [33,34].

Only a limited number of studies have previously investigated the risk factors associated with FL [22,35,36]. In concordance with prior studies, multivariate analysis reveals the Ann Arbor stage as a significant predictor of OS and LSS in FL patients, emphasizing the critical role of early diagnosis, indicating the importance of early detection [37,38]. Moreover, the data demonstrated a decline in patient survival with older ages. This condition may be associated with immunological insufficiency and somatic decline in elderly people. Studies have revealed that individuals with B symptoms have a poor prognosis, emphasizing the need for careful monitoring of high-risk patients [39].

Notably, in the multivariate analysis of OS, several covariates demonstrated statistically significant hazard ratios (HRs) with confidence intervals (CIs) approaching 1.00. Specifically, higher income, nonmetropolitan residence, Grade 3 histology, and SPMs showed borderline effect sizes. While statistically significant, the narrow deviation of these HRs from 1.00 suggests limited clinical impact. This phenomenon may reflect either intrinsic biological effects (e.g., socioeconomic factors like income and geography exerting subtle systemic influences) or methodological constraints (e.g., inability to distinguish Grade 3A from 3B in registry data, potentially diluting prognostic discrimination). Notably, the delayed OS impact of SPMs at 91 months in Kaplan-Meier analysis further implies temporal effect modification. Similar patterns emerged in LSS analysis, where non-Hispanic ethnicity, higher income, and Grade 3 demonstrated marginal associations. These findings warrant cautious interpretation: although meeting statistical significance thresholds, the clinical relevance of such small-magnitude effects requires validation through risk-stratified interventions or mechanistic studies. Future studies should incorporate detailed histopathological subclassification and longitudinal socioeconomic metrics to improve prognostic precision.

Our survival analysis revealed distinct therapeutic impacts on patient outcomes. Initial chemotherapy emerged as a significant risk factor for diminished LSS, whereas it showed no measurable impact on OS. In contrast, both radiotherapy and surgical intervention demonstrated significant associations with reduced all-cause mortality in FL patients. Notably, among patients developing SPMs, initial radiotherapy maintained protective effects for both OS and LSS. These findings emphasize the necessity for personalized therapeutic decision-making that carefully weighs these differential survival outcomes. The consistency of these findings with current clinical guidelines reinforces the validity of our study conclusions [2].

Our study has limitations: First, the retrospective SEER-based design introduces potential selection bias through uneven population representation, possibly compromising the generalizability of survival outcomes (particularly OS/LSS) beyond U.S. healthcare contexts. Second, absent granular treatment data (radiation fields, chemotherapy regimens/ schedules, surgical approaches) and FLIPI scores prevents adjustment for critical prognostic confounders, potentially distorting therapeutic effect estimates. The inability to distinguish histological grades 3A vs 3B—subtypes with divergent clinical behaviors—may have introduced misclassification bias, obscuring true survival differences. While we conducted rigorous multivariable adjustments, these data gaps likely introduced residual confounding. Future studies should use prospective data to validate and expand these findings.

## 5. Conclusion

In conclusion, using the largest FL cohort to date, our study meticulously describes the epidemiological characteristics of FL. After excluding the potential for transformation to more aggressive subtypes and the competing risk of death, the study identifies the risk factors contributing to the development of SPMs in FL patients and delves into the causes of death among patients. Additionally, we analyzed the risk factors affecting the survival rates of FL patients, as well as those patients with SPMs. These results carry translational significance by informing risk-adapted surveillance strategies – particularly the need for enhanced malignancy screening protocols in high-risk subgroups and personalized comorbidity

management in elderly populations. Future investigations should focus on validating these predictive models across diverse healthcare settings, elucidating the molecular mechanisms underlying SPM pathogenesis in FL, and evaluating the clinical utility of targeted screening interventions through prospective trials.

## Supporting information

**S1 Fig. Prevalence of follicular lymphoma.** (A)Prevalence of follicular lymphoma by gender (B)Prevalence of follicular lymphoma by age (C) per 10,000: Surveillance, Epidemiology, and End Results (SEER) 2000–2020.
(TIF)

**S1 Table. Follicular lymphoma prevalence with different characteristics per 100,000: Surveillance, Epidemiology, and End Results (SEER) 2000–2020.**
(DOCX)

**S2 Table. Follicular lymphoma treatment modalities.**
(DOCX)

**S3 Table. Primary locations of patients with follicular lymphoma at diagnosis.**
(DOCX)

**S4 Table. Second primary malignancies in follicular lymphoma.**
(DOCX)

**S5 Table. Comparison of baseline characteristics between patients with and without deleted Second Primary Malignancies occurring within 6 months.**
(DOCX)

**S6 Table. Univariate competing risks and Cox regression analysis of risk factors for Second Primary Malignancies occurrence.**
(DOCX)

**S7 Table. Multivariable competing risks and Cox regression analysis of risk factors for Second Primary Malignancies occurrence.**
(DOCX)

**S8 Table. Univariate competing risks and Cox regression analysis of risk factors for lymphoma-specific survival.**
(DOCX)

**S9 Table. Multivariable competing risks and Cox regression analysis of risk factors for lymphoma-specific survival.**
(DOCX)

**S10 Table. Univariate Cox regression analysis of risk factors for overall survival.**
(DOCX)

**S11 Table. Multivariable Cox regression analysis of risk factors for overall survival.**
(DOCX)

**S12 Table. Univariate Cox regression analysis of risk factors for overall survival in Second Primary Malignancies patients.**
(DOCX)

**S13 Table. Multivariable Cox regression analysis of risk factors for overall survival in Second Primary Malignancies patients.**
(DOCX)

**S14 Table. Univariate competing risks and Cox regression analysis of risk factors for lymphoma-specific survival in Second Primary Malignancies patients.**
(DOCX)

**S15 Table. Multivariable competing risks and Cox regression analysis of risk factors for lymphoma-specific survival in Second Primary Malignancies patients.**
(DOCX)

**S16 Table. Dataset1(including patients with SPMs occurring within less than 6 months from diagnosis).**
(XLSX)

**S17 Table. Dataset2(excluding patients with SPMs occurring within less than 6 months from diagnosis).**
(XLSX)

## Acknowledgments

The authors thank all the study patients, investigators, and coordinators.

## Author contributions

**Conceptualization:** Caigang Xu.

**Data curation:** Ying Tian, Wanxi Yang.

**Methodology:** Yuanxiao Li, Wenjiao Tang.

**Resources:** Juan Xu, Wenjiao Tang, Caigang Xu.

**Software:** Ying Tian.

**Supervision:** Juan Xu, Caigang Xu.

**Writing – original draft:** Ying Tian, Wanxi Yang, Juan Xu, Yuanxiao Li.

**Writing – review & editing:** Caigang Xu.

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
