## [Decision Letter · Decision Letter 0]

12 Mar 2025

PONE-D-25-03313Risk and prognosis of second primary malignancies in patients with follicular lymphoma in the era of rituximab: a population study based on the SEER databasePLOS ONE

Dear Dr. Xu,

Thank you for submitting your manuscript to PLOS ONE. After careful consideration, we feel that it has merit but does not fully meet PLOS ONE’s publication criteria as it currently stands. Therefore, we invite you to submit a revised version of the manuscript that addresses the points raised during the review process.

The manuscript presents valuable research but needs significant improvements in clarity, methodology, discussion, and statistical reporting before it can be considered for publication. 

We look forward to receiving your revised manuscript.

Kind regards,

M Tanveer Hossain Parash, MBBS, M Phil

Academic Editor

PLOS ONE

Additional Editor Comments:

Line 50 Introduction: It would benefit from a more detailed discussion of the current gaps in the literature, particularly regarding the impact of rituximab on SPM risk. The authors should also clarify why their study is necessary and how it advances the field beyond previous studies.

Line 73 Methods: Please add study design and period. Please write the inclusion criteria. Further justification is needed for excluding SPMs diagnosed within six months of FL diagnosis. Were sensitivity analyses conducted to validate this choice? Please mention sample size calculations and sampling method.

Line 74-79: The authors mention that ethical approval was waived due to the use of de-identified data. However, they should provide more details on how they ensured the ethical use of the data, including any specific guidelines or regulations they followed.

Line 80-92: The authors should provide more details on the SEER database, including its limitations and potential biases. For example, how representative is the SEER database of the general population, and are there any known biases in the data collection process? The absence of FLIPI scores, genetic markers, and detailed treatment regimens should be acknowledged more explicitly.

Line 107-113: The study objectives should be summarized and included as the final paragraph of the introduction.

Line 114-135: The authors mention using competing-risk regression analysis, but they should provide more details on the specific models used and how they handled competing risks. Additionally, they should clarify how they addressed potential confounding factors in their analysis.

Line 194- 199: Table 1: The table provides a good summary of patient characteristics, but it would be helpful to include more details on the treatment modalities (e.g., specific chemotherapy regimens, radiation doses) and how they were categorized.

Figure 1 and 2: The figures are informative, but the authors should ensure that all abbreviations are defined in the figure legends. Additionally, they should consider adding confidence intervals to the forest plot in Figure 2 to provide a more complete picture of the results.

Line 278-287: The authors should provide more details on the survival analysis, including the specific time points used for the Kaplan-Meier curves and how they handled censoring in their analysis.

Line 311: Discussion:

The authors should provide a more detailed comparison of their findings with previous studies, particularly regarding the risk factors for SPMs and the impact of rituximab therapy. They should also discuss any discrepancies between their findings and those of previous studies. For example, the observation that Grade 3 FL patients had a lower risk of developing SPMs contradicts previous studies. A more thorough exploration of potential biological mechanisms is needed.

The authors should expand on the clinical implications of their findings, particularly regarding the management of FL patients at risk for SPMs. They should also discuss any potential strategies for reducing the risk of SPMs in this population.

Some hazard ratios (HRs) have confidence intervals (CIs) that approach 1.00. The clinical significance of these borderline findings should be addressed.

The potential impact of changing treatment strategies over time on survival outcomes should be further discussed.

The authors acknowledge some limitations of their study, but they should provide a more detailed discussion of these limitations and how they might have affected the results. For example, they should discuss the potential impact of missing data, selection bias, and the lack of information on specific treatment regimens.

The conclusion is concise and summarizes the main findings of the study. However, the authors should consider adding a brief discussion of the broader implications of their findings for future research and clinical practice

Reviewers' comments:

Reviewer's Responses to Questions

**Comments to the Author**

1. Is the manuscript technically sound, and do the data support the conclusions?

Reviewer #1: Yes

Reviewer #2: Yes

2. Has the statistical analysis been performed appropriately and rigorously? 

Reviewer #1: Yes

Reviewer #2: Yes

3. Have the authors made all data underlying the findings in their manuscript fully available?

Reviewer #1: Yes

Reviewer #2: Yes

4. Is the manuscript presented in an intelligible fashion and written in standard English?

Reviewer #1: Yes

Reviewer #2: Yes

5. Review Comments to the Author

Reviewer #1: Overall, the manuscript was well written and comprehensively organized. The abstract, along with other components in the manuscript were well addressed and reasonable to the title and its objectives. The tables and figures provided were also helpful to understand the discussed topic. The manuscripts addressed the second primary malignancies in patients with follicular carcinoma using SEER database. There are several comments that the authors need to improvised in order for the manuscript to more understandable. For example:

ABSTRACT

The authors should address the conclusion in the abstract as it can summarize the author’s overall findings dan the future plan of the study.

INTRODUCTION

It would be more beneficial if the authors could elaborate more in regards to the mortality rate of FL in recent years, the global prevalence of FL and how it sparks your interest to conduct this retrospective study.

METHODS

Is it possible for the authors to mention the specific date of the data were taken, as I can see the year 2000 to 2020 is too general.

RESULTS

As the years of FL incidence were reported at the first 2 paragraphs of the result section (Line 138-148), I would advise the authors to summarized it as well in a simple line graph for the readers to understand in more easily.

DISCUSSION

Discussion part were presented quite well and easy to be understood. In addition to this, I would suggest that the author to add 1-2 more sentences to explain the strength of the current study.

CONCLUSION

I would suggest that the conclusion section to be separated from the discussion section.

I wish you all the best to correct the manuscript and I am looking forward to review the revised version of this manuscript.

Reviewer #2: Well written manuscript. Just a small grammatical error correction needed. It is also suggested that the author updated the list of references (at least in the last five years).

The reviewer could not comment much on the content as this is not his area of expertise.

6. PLOS authors have the option to publish the peer review history of their article (what does this mean? ). If published, this will include your full peer review and any attached files.

**Do you want your identity to be public for this peer review?** For information about this choice, including consent withdrawal, please see our Privacy Policy .

Reviewer #1: No

Reviewer #2: No

---

## [Author Response · Author response to Decision Letter 1]

24 Apr 2025

Dear Editor and Esteemed Reviewers,

We sincerely appreciate your valuable insights and constructive suggestions on our manuscript. All raised concerns have been systematically addressed through the following point-by-point responses: each query has been bolded and numbered sequentially for efficient navigation, critical revisions are highlighted in red font for instant visual identification.

Should any clarification or additional information be required, please contact us at xucaigang@wchscu.cn. We remain fully available for further modifications as needed.

We are profoundly grateful for your expert guidance in enhancing the scientific rigor of this work, and respectfully await your further instruction.

Sincerely,

Caigang Xu

Corresponding Author

Additional Editor Comments:

1.Line 50 Introduction: It would benefit from a more detailed discussion of the current gaps in the literature, particularly regarding the impact of rituximab on SPM risk. The authors should also clarify why their study is necessary and how it advances the field beyond previous studies.

We sincerely appreciate the editor's insightful comments. In accordance with the suggestions, we have substantially revised the Introduction section, with key modifications highlighted in red to explicitly address the editorial concerns and enhance the clarity of our responses. These revisions have been consistently incorporated throughout the manuscript to ensure uniformity with the updated content.

Introduction

Follicular lymphoma (FL), originating from the germinal center B cells in lymph nodes or lymphoid tissue, is one of the most common indolent forms of non-Hodgkin's lymphoma (NHL). In Western countries, FL accounts for about 20-35% of NHL cases[1, 2], with an age-standardized incidence of 2–4 per 100,000 person-years[1]. The therapeutic landscape for FL has been revolutionized by anti-CD20 monoclonal antibodies, particularly rituximab-based immunochemotherapy, which has elevated 10-year overall survival (OS) rates beyond 80%[2]. While these advancements have substantially improved survival outcomes, emerging survivorship challenges including long-term treatment toxicities and second primary malignancies (SPMs) demand increased clinical attention. Notably, contemporary studies indicate that SPM incidence remains unaffected by the incorporation of rituximab into therapeutic regimens[3-6], emphasizing the critical need for optimized surveillance strategies and preventive measures in the current era of rituximab-dominated therapies.

Previous studies have showed that FL patients face a higher SPMs risk than the general population, with factors such as older age, male, chemotherapy, radiotherapy, radioimmunotherapy, autologous stem cell transplantation following high-dose chemotherapy, multiple treatments, and B symptoms linked to SPM development[2, 5, 7-10]. However, existing research presents conflicting conclusions regarding risk factor associations, compounded by methodological limitations: (1) most analysis predate widespread rituximab adoption, potentially obscuring modern risk profiles shaped by newer combination therapies (e.g., rituximab plus lenalidomide) [11]; (2) conventional Kaplan-Meier analysis overestimate SPM incidence by neglecting competing mortality risks; and (3) the clinical characteristics, survival outcomes, and prognostic determinants for FL patients post-SPM diagnosis remain poorly characterized due to SPM rarity and insufficient longitudinal data.

To address these knowledge gaps, we conducted a population-based cohort study with three principal objectives: First, to identify clinical predictors of SPM development in FL patients treated during the rituximab era (post-2000). Second, to identify distinct prognostic determinants that differentiate OS from lymphoma-specific survival (LSS) between FL patients and the subgroup developing secondary malignancies following FL diagnosis. Third, to delineate cause-specific mortality patterns distinguishing these patient subgroups. Methodologically, we employ Fine-Gray competing risk models to provide more accurate SPM risk estimation compared to traditional survival analysis. Our findings aim to inform risk-adapted surveillance protocols and refine therapeutic decision-making in FL management.

2.Line 73 Methods: Please add study design and period. Please write the inclusion criteria. Further justification is needed for excluding SPMs diagnosed within six months of FL diagnosis. Were sensitivity analyses conducted to validate this choice? Please mention sample size calculations and sampling method.

2.2 Patient selection

This retrospective cohort study analyzed clinical data and follow-up information from patients with follicular lymphoma from 2000 to 2020. Inclusion criteria were (1) first primary FL with histologically confirmed (ICD-O-3 histological codes 9690-9698); (2) age>14 years; (3) complete and reliable follow-up data. The exclusion criteria included (1)did not have complete and reliable clinical characteristics; (2)diagnosis established post-mortem; (3)SPMs diagnosed within a six-month interval following the initial FL diagnosis; (4)SPMs of B-cell lineage, or documented relapse of FL; (5) survival time was 0. Notably, our sensitivity analysis included all SPM cases regardless of diagnostic interval(Table S5-15).The SPM definition followed National Cancer Institute guidelines as described by Morris et al[12], requiring ≥6-month interval between FL diagnosis and subsequent malignancy. As a population-based registry study utilizing the SEER-17 dataset (2000-2020), our analysis included all eligible follicular lymphoma cases meeting predefined criteria. This exhaustive sampling approach eliminates selection bias inherent to calculated sample sizes, ensuring maximal representation of the US population covered by SEER registries (covers approximately 26.5% of the U.S. population). The aforementioned text has been incorporated into the manuscript at Line 147-166.

3.Line 74-79: The authors mention that ethical approval was waived due to the use of de-identified data. However, they should provide more details on how they ensured the ethical use of the data, including any specific guidelines or regulations they followed.

The data is public and patient information is anonymous, so our study does not require ethical approval or informed consent of patients. Our research follows the regulations published by the SEER database and the Declaration of Helsinki. The aforementioned text has been incorporated into the manuscript at Line 143-146.

4.Line 80-92: The authors should provide more details on the SEER database, including its limitations and potential biases. For example, how representative is the SEER database of the general population, and are there any known biases in the data collection process? The absence of FLIPI scores, genetic markers, and detailed treatment regimens should be acknowledged more explicitly.

This study extracted de-identified patient-level records from the SEER Research Data (17 Registries, November 2022 Submission, 2000-2020) using SEER*Stat 8.4.3. While the SEER-17 registries collectively cover approximately 26.5% of the U.S. population, their geographic distribution requires careful interpretation: The participating regions predominantly consist of selected states [California (excluding San Francisco/San Jose-Monterey/Los Angeles areas)�Connecticut, Hawaii, Iowa, Kentucky, Louisiana, New Jersey, New Mexico, Utah] and metropolitan areas (San Francisco-Oakland, Los Angeles, Seattle-Puget Sound, Atlanta, Greater Georgia, Rural Georgia, San Jose-Monterey), with intentional oversampling of specific populations (e.g., Alaska Natives). This deliberate sampling design may introduce geographic and demographic imbalances, particularly underrepresenting rural populations in non-SEER states. Three key limitations merit explicit acknowledgment. First, the absence of follicular lymphoma international prognostic index (FLIPI) scores prevents risk stratification, potentially masking survival differences among biological subgroups. Second, the absence of documented genetic biomarkers (e.g., BCL2 expression status, EZH2 mutation profiles, and chromosomal translocation patterns) hinders comprehensive assessment of genomic heterogeneity's prognostic implications. Third, while therapeutic modalities (chemotherapy, radiotherapy, and surgical resection) are recorded as binary variables, the lack of detailed treatment specifications including protocol variations, dose-intensity metrics, and temporal coordination of interventions may introduce confounding bias in comparative effectiveness research. The aforementioned text has been incorporated into the manuscript at Line 118-142.

5.Line 107-113: The study objectives should be summarized and included as the final paragraph of the introduction.

In response to the editor's comment, we have refined the research objectives and incorporated the updated summary into the Introduction (Lines [105-110]).

In this population-based cohort study, we aimed to (1) identify clinical risk factors for SPMs in FL patients, (2) establish prognostic predictors for both overall survival (OS) and lymphoma-specific survival (LSS) comparing FL patients with versus without SPMs, and (3) delineate differential patterns of cause-specific mortality between these two clinically distinct populations.

6.Line 114-135: The authors mention using competing-risk regression analysis, but they should provide more details on the specific models used and how they handled competing risks. Additionally, they should clarify how they addressed potential confounding factors in their analysis.

We employed the Fine-Gray proportional subdistribution hazards model as the primary analytical framework to quantify associations between covariates and outcomes while accounting for the interdependence of competing events. Specifically, two distinct competing-risk regression analysis were conducted: (1) For SPMs, the primary event was defined as time from FL diagnosis to SPM detection, with all-cause mortality treated as the competing event; (2) For lymphoma-specific survival, the primary event was death from FL, with non-FL-related deaths constituting the competing event. Patients were censored at last follow-up if neither event occurred. Cumulative incidence functions were estimated using the Aalen-Johansen estimator to avoid overestimation biases inherent in Kaplan-Meier methodology. To address confounding, we implemented a rigorous covariate selection protocol: first, clinically plausible variables were pre-specified through literature review; second, univariate screening with an inclusive threshold (P≤0.2) retained marginally significant factors; finally, multivariable models incorporated these predictors alongside established prognostic variables. Sensitivity analysis evaluated model robustness through subgroup stratification by SPM latency periods (≤6 vs. >6 months post-diagnosis) and methodological cross-validation using conventional Cox proportional hazards models, confirming the stability of identified risk factors across analytical paradigms(Table S5-15). The aforementioned text has been incorporated into the manuscript at Line 190-211.

7.Line 194- 199: Table 1: The table provides a good summary of patient characteristics, but it would be helpful to include more details on the treatment modalities (e.g., specific chemotherapy regimens, radiation doses) and how they were categorized.

We sincerely appreciate the editor's valuable insights regarding treatment modality specifications. As detailed in our Methods section, the SEER database presents inherent limitations in chemotherapy documentation, restricting classification to a binary system (Yes/No or Unknown) without details regarding specific therapeutic regimens. For radiotherapy parameters, while the original dataset contained eight categorical classifications [including: Beam radiation, Radioactive implants (brachytherapy; 1988+), Radioisotopes (1988+), Combination therapies, Radiation NOS (not otherwise specified), Recommended (unknown administration status), Refused (1988+), and None/Unknown], we intentionally consolidated these into clinically interpretable binary variables (Yes/No or Unknown) to facilitate meaningful analysis.

We fully concur with the editor's recognition of the critical importance of treatment particulars such as radiation dosimetry parameters and chemotherapy protocol specifics in outcome evaluation. However, it is essential to emphasize that these operational-level treatment details fundamentally exceed the current documentation scope of the SEER registry infrastructure. This structural limitation inherently constrains the therapeutic information available for secondary analysis, while maintaining the database's strengths in population-level observational research.

8.Figure 1 and 2: The figures are informative, but the authors should ensure that all abbreviations are defined in the figure legends. Additionally, they should consider adding confidence intervals to the forest plot in Figure 2 to provide a more complete picture of the results

We appreciate the reviewer's insightful suggestion. Here's the revised figure legends section with modifications to address the editor's comments:

Revised Figure Legends

Figure 1. Cumulative incidence of second primary malignancies in patients with follicular lymphoma diagnosed between 2000 and 2020.

Abbreviations: TTT, time to treatment; NHL, non-Hodgkin's lymphoma; Grade NOS, Not Otherwise Specified.

Race categories: "Others" includes American Indian/AK Native and Asian/Pacific Islander.

Marital status categories: "Others" includes divorced, separated, unmarried/domestic partner, and widowed.

Figure 2. Forest plot of multivariate competing risks regression analysis for risks of second primary malignancies in follicular lymphoma patients.

Abbreviations: NHL, non-Hodgkin's lymphoma; Grade NOS, Not Otherwise Specified.

Race categories: "Others" includes American Indian/AK Native and Asian/Pacific Islander.

Marital status categories: "Others" includes divorced, separated, unmarried/domestic partner, and widowed.

Figure 3. Survival analysis of follicular lymphoma patients based on secondary primary malignancy status.

(A) Overall survival (OS) stratified by occurrence of secondary primary malignancies (SPMs);

(B) Lymphoma-specific survival (LSS) stratified by occurrence of SPMs;

(C) Log-rank test comparison of OS between patients with and without SPMs;

(D) Log-rank test comparison of LSS between patients with and without SPMs.

These changes ensure each figure legend is self-contained, abbreviations are properly defined, and statistical completeness is maintained with the addition of confidence intervals in Figure 2.

9.Line 278-287: The authors should provide more details on the survival analysis, including the specific time points used for the Kaplan-Meier curves and how they handled censoring in their analysis.

In accordance with the editor's valuable recommendations, we have implemented substantive revisions to optimize the section's organizational structure and analytical precision. The key modifications include: (1) Integration of original Sections 3.4-3.5 into a cohesive Survival Analysis framework to eliminate redundancy; (2) Strategic expansion with two specialized subsections: 3.5 Prognostic factors among FL patients;3.6 Prognostic factors among FL patients developing SPMs.

We executed comprehensive survival analysis adhering to the following methodological standards:

Survival Probabilities:

Kaplan-Meier curves were specifically calibrated to capture survival rates at four clinically pivotal intervals (1-, 3-, 5-, and 10-year landmarks), reflecting both the indolent biological behavior of FL and standardized clinical surveillance protocols.

Censoring Protocol:

A rigorous right-censoring algorithm was implemented at the earliest occurrence of (1) Study closure (December 31, 2020); (2)Last confirmed alive date for lost-to-follow-up patients; (3)Mortality events unrelated to FL (exclusively for lymphoma-specific survival calculations)

Endpoint Operationalization:

Overall Survival (OS): Defined by all-cause mortality.

Lymphoma-Specific Survival (LSS): Defined as deaths

---

## [Editor Report · Decision Letter 1]

27 Apr 2025

Risk and prognosis of second primary malignancies in patients with follicular lymphoma in the era of rituximab: a population study based on the SEER database

PONE-D-25-03313R1

Dear Dr. Xu,

We’re pleased to inform you that your manuscript has been judged scientifically suitable for publication and will be formally accepted for publication once it meets all outstanding technical requirements.

Kind regards,

M Tanveer Hossain Parash, MBBS, M Phil

Academic Editor

PLOS ONE
---

## [Editor Report · Acceptance letter]

PONE-D-25-03313R1

PLOS ONE

Dear Dr. Xu,

I'm pleased to inform you that your manuscript has been deemed suitable for publication in PLOS ONE. Congratulations! Your manuscript is now being handed over to our production team.

Kind regards,

on behalf of

Dr. M Tanveer Hossain Parash

Academic Editor

PLOS ONE